# GAN-Based Image Colorization for Self-Supervised Visual Feature Learning

**DOI:** 10.3390/s22041599

**Published:** 2022-02-18

**Authors:** Sandra Treneska, Eftim Zdravevski, Ivan Miguel Pires, Petre Lameski, Sonja Gievska

**Affiliations:** 1Faculty of Computer Science and Engineering, University Ss. Cyril and Methodius, 1000 Skopje, North Macedonia; eftim.zdravevski@finki.ukim.mk (E.Z.); petre.lameski@finki.ukim.mk (P.L.); sonja.gievska@finki.ukim.mk (S.G.); 2Instituto de Telecomunicações, Universidade da Beira Interior, 6200-001 Covilhã, Portugal; impires@it.ubi.pt; 3Escola de Ciências e Tecnologias, University of Trás-os-Montes e Alto Douro, Quinta de Prados, 5001-801 Vila Real, Portugal

**Keywords:** self-supervised learning, transfer learning, image colorization, convolutional neural network, generative adversarial network

## Abstract

Large-scale labeled datasets are generally necessary for successfully training a deep neural network in the computer vision domain. In order to avoid the costly and tedious work of manually annotating image datasets, self-supervised learning methods have been proposed to learn general visual features automatically. In this paper, we first focus on image colorization with generative adversarial networks (GANs) because of their ability to generate the most realistic colorization results. Then, via transfer learning, we use this as a proxy task for visual understanding. Particularly, we propose to use conditional GANs (cGANs) for image colorization and transfer the gained knowledge to two other downstream tasks, namely, multilabel image classification and semantic segmentation. This is the first time that GANs have been used for self-supervised feature learning through image colorization. Through extensive experiments with the COCO and Pascal datasets, we show an increase of 5% for the classification task and 2.5% for the segmentation task. This demonstrates that image colorization with conditional GANs can boost other downstream tasks’ performance without the need for manual annotation.

## 1. Introduction

It has been shown that deep convolutional neural networks (CNNs) have a great ability to learn features from visual data. Because of this, they have been consistently used as a base for many computer vision tasks such as image classification [1], object detection [2,3], semantic and instance segmentation [4,5], image captioning [6], and more. The success of deep learning models greatly depends on the amount of data used for their training. They have a capacity for scaling up and increasing in complexity with more training data.

Large-scale image datasets such as ImageNet [7], COCO [8], Pascal VOC [9], Places [10], and others, have been proposed for training deep convolutional networks. However, collecting and annotating these large datasets consisting of millions of images is hard, expensive, and time-consuming. Moreover, it is prone to human errors and is also limited in building models for new domains. For example, annotating medical images requires the contribution of an expert in the field. Another example domain is image segmentation, where creating annotations is especially tedious manual work.

The traditional method to use when a large annotated dataset is not available is to use transfer learning [11]. Usually, models with general knowledge trained on ImageNet are used as pretrained models and are then fine-tuned for specific tasks. This process provides a good starting point for most convolutional models, as they come with already learned basic features. However, this approach is still limited to the general knowledge gained from ImageNet and, therefore, may not be helpful for specific domains [12].

To avoid these problems, self-supervised feature learning methods have been proposed, which learn from images that have not been manually annotated [13]. These models are based on convolutional neural networks that learn visual features through solving a pretext task. A number of pretext tasks have been proposed such as image inpainting [14], super-resolution [15] and image colorization [16]. The benefit of using these approaches come from the fact that the pseudolabels needed for the pretext task can be automatically generated. Therefore, any manual human effort is avoided. Models use raw, unlabeled data and produce meaningful representations that can be transferred to other tasks.

This paper focuses on exploring image colorization as a pretext task for visual feature learning. Image colorization is the process of adding color to an originally black and white image. It falls in the category of self-supervised tasks because any image can be split into its gray and color components. For example, when working in the Lab color space, the L-channel is used as input to the image colorization model, while the a and b channels are the pseudo labels that the model should learn to predict.

In this paper, first, we implement a conditional generative adversarial network (cGAN) [17] and train it on the pretext task of image colorization. Our main contribution is the usage of a generative adversarial network for the first time for the purpose of self-supervised visual feature learning through image colorization. We choose a GAN as a base for our model, as GANs have been proven to yield the most realistic results for the image colorization problem. After the model is trained, the learned visual features are transferred to two downstream tasks, multilabel image classification [18] and semantic segmentation [19]. The goal is to explore whether the performance of these downstream tasks can be improved using image colorization pretraining. Through the task of image colorization, the generative adversarial network learns visual features about the objects present on the training images, such as color, form, and texture. The rationale is that these visual features are general and could be useful to any other computer vision task, starting from image classification, object detection, and segmentation to other more complex tasks from the domain of image generation. We chose to apply image colorization on image classification and segmentation as downstream tasks, as they lay the foundations for most computer vision models.

The remainder of the paper is structured as follows. Section 2 reviews some state-of-the-art related works. Then, in Section 3, we review the proposed methods for image colorization and elaborate the training process. In Section 4, we review the used datasets, architecture, experimental setup, and achieved results. In Section 5, we discuss the results and point out directions for future work. Finally, Section 6 concludes the paper.

## 2. Related Work

Visual self-supervision has been a field of great interest for researchers in the past few years. Self-supervised visual feature learning methods can be grouped into four categories based on the used pretext task: context-based, free-semantic label-based, cross-modal-based, and generation-based [13]. Context-based pretext tasks are designed based on the context similarity between image patches [20], the spatial relationship among patches [21], or based on temporal frame order from videos [22]. Free semantic label-based methods include pretext tasks such as moving object segmentation [23], contour detection [24], and relative depth prediction [24]. They use pseudolabels generated by game engines or hard-coded algorithms. Cross-modal-based methods use pretext tasks to verify if two input data channels are corresponding to each other. Methods that belong in this group are egomotion [25], RGB-flow correspondence verification [26] and visual–audio correspondence verification [27].

The generation-based methods are the most related class of self-supervised visual feature learning methods to our research. These methods learn visual features by solving pretext tasks that require the generation of images. One of these methods is image inpainting, which predicts missing regions from an image. For this task, the input is automatically generated by removing one area from an image, while the target is the whole image itself. To infer the missing regions, a model needs to learn the color and structure of common objects. This knowledge is gained using generative adversarial networks (GANs), which learn the semantic features of images that can then be transferred to other similar tasks. Only the approach described in [14] analyzes the advantages of transferring features learned by image inpainting to other tasks, and the authors have achieved competitive results with other models that use supervised pretraining. Authors of [28] show that self-supervised learning can be used for anomaly detection by learning deep representations and then building a generative one-class classifier on learned representations. Another approach described in [29] proposes a deraining method using an unsupervised deraining generative adversarial network that resolves issues of previous approaches by introducing self-supervised constraints from the intrinsic statistics of unpaired rainy and clean images. Another generation-based method for learning visual features is super-resolution [15]. This is the task of transforming a low-resolution image into a high-resolution image. With the help of generative adversarial networks, fine features are learned from training images, resulting in realistic yet generated images. Unfortunately, to the best of our knowledge, the potential benefits of transferring features learned by super-resolution to other tasks have not been studied yet. Some example self-supervised methods for learning visual features are shown in Figure 1.

Finally, the topic of this research—image colorization as a proxy task for visual understanding—was first proposed as a part of two automatic colorization methods described in [16,30]. Authors of [16] train an AlexNet model for the colorization task, using the ImageNet dataset and then fine-tune the network on the PASCAL dataset for the tasks of classification, detection, and segmentation. They compared the results to other self-supervised methods and found that their approach achieves higher performance on object classification and segmentation. However, the authors have also stated that all self-supervised methods are still behind the traditional Imagenet supervised pretraining. Another important finding is that training on color input images, as opposed to gray ones, makes almost no difference in the performance of the models. Authors of [30] train a VGG-16 model using image colorization as a proxy task and then fine-tune the model for PASCAL segmentation. They report state-of-the-art results for segmentation task compared to other self-supervised methods.

The first in-depth analysis for visual feature learning by image colorization is provided in [31]. Network details, such as loss, receptive field, batch normalization, and padding, are also investigated. Different model architectures are tried out, including AlexNet, VGG-16, and ResNet-152. The models are pretrained on the ImageNet and Places datasets and fine-tuned on PASCAL VOC classification and segmentation. The results show that the ResNet-152 model has a better performance on both tasks than other self-supervised methods. This suggests that self-supervised image colorization can benefit from a high complexity network.

Another paper that investigates self-supervised representation learning is [32]. The authors propose an architecture that modifies the traditional autoencoder by adding a split to the network that results in two disjoint subnetworks. For the task of image colorization, one network predicts the color channels given the grayscale channel. In contrast, the other network predicts the grayscale channel provided the color channels. This way, the model extracts features from the entire input signal. The base of the network uses AlexNet, pretrained on ImageNet and Places datasets, and fine-tuned on PASCAL classification, segmentation, and detection. Comparison of previous research papers that investigate image colorization as a proxy task is provided in Table 1. Their best results on different downstream tasks are shown in Table 2.

All previous research has relied on convolutional neural networks to extract features from an image. The most prevalent architecture is AlexNet, used for easier comparison with other self-supervised methods, which are also based on AlexNet. However, this paper explores the use of generative adversarial networks for image colorization as a proxy task for the first time. Generative adversarial networks have been proven to yield more realistic colorization results [33,34,35] than simple convolutional neural networks. Therefore, we expect them to learn a better feature representation that can be then transferred to other tasks. In addition, as stated by [31], image colorization as a proxy task should benefit from an even more complex network than a CNN, and an example of such a network is a generative adversarial network.

## 3. Methods

In this section, we introduce image colorization as a target task, provide an overview of the architecture of generative adversarial networks used for colorization, and explain their learning procedure. The data flow diagram of the proposed use of the colorization model and transferring its weights on the subsequent tasks for multiclass classification and region segmentation are shown in Figure 2.

### 3.1. Image Colorization

Image colorization is the process of converting grayscale images to their colorful versions. Digitally, this is done by using luminance–chrominance color spaces that allow us to separate the pixel intensity information from the pixel color information. Image colorization is still an active area of research. Starting from semiautomatic approaches that involved using reference images to extract color [36], or a user to give hints to an algorithm [37], today, fully automatic methods are more widespread. Fully automatic approaches utilize the advances in the computer vision area of deep learning, ranging from deep convolutional networks to GANs. These models are trained on large image datasets and automatically learn a mapping from gray to color pixels.

A great number of models based on convolutional neural networks have been proposed over the years [16,30,38,39]. CNNs are an appropriate choice for an image colorization network because they can retain an image’s spatial and temporal dependencies. Models that formulate the problem as regression [38,39] try to minimize the mean squared errors between the original and predicted color pixels. However, the usage of L2 loss encourages conservative predictions, and the generated images result in muted colors. Other papers [16,30] treat the problem as classification and predict a distribution of possible colors for each pixel. This approach improves the vibrancy and diversity of the generated colors.

The majority of new papers researching image colorization involve the usage of a GANs. What makes GANs excel in the image colorization domain is their ability to learn an appropriate loss function alongside a mapping function. For example, in [40], authors show that GANs can be used to generate data of the minority class and, as a result, boost the classification accuracy in problems with heavy class imbalance. In the next subsection, we explain generative adversarial networks for image colorization in more detail.

### 3.2. Generative Adversarial Networks for Image Colorization

Generative adversarial networks are generative models composed of two opposing parts—a generator and a discriminator. The task of the generator is to produce outputs that are indistinguishable from reality, while the task of the discriminator is to differentiate between the real and generated images. In addition to the adversarial loss, most models also use L1 loss, which forces the generator to produce results that are structurally similar to the ground truth images. Because of this reason, most image colorization research papers have been exploring the benefits of GANs.

Conditional GANs are the most suitable for the problem of image colorization, as they need to condition the network on a grayscale input image and generate a color output image. The first paper that investigates the usage of cGANs for image colorization is [33], called Pix2Pix. This model provides a general solution to a family of paired image-to-image translation problems. The goal is to map two domains, such as grayscale to color. The generator used is a U-Net, which progressively downsamples the image, until a bottleneck, after which the process is reversed, and the image is upsampled to its original size. Skip connections are also added to facilitate the flow of low-level information through the network. The discriminator used is called a PatchGAN, whose job is to decide whether each image patch is real or fake. An essential part of the model is the addition of dropout layers, which help add diversity to the results.

The approach described in [34] aims to extend Pix2Pix, by generalizing it to high-resolution images. It suggests training strategies to speed up and stabilize the training process. Authors of [35] propose a model that differs from Pix2Pix. In particular, there is no downsizing of the input image, but more importantly, a multilayer concatenation of the grayscale image is performed continuously to maintain reality. The discriminator used by [35] is simpler, as it is a binary classifier predicting only one label, real or fake, for the entire image. The generator by [36] is a VGG-16 pretrained network, but instead of using the output of the last layer, they stack all activations above each pixel to form hypercolumns, which are further used for prediction. Authors of [41] explore modifying the discriminator by using a capsule network instead of a traditionally used convolutional network. The approach described in [42] utilizes a similar generator architecture as the networks used in [38,39], while the discriminator is a PatchGAN. The authors also experiment with the loss function and employ WGAN loss alongside L2 loss for color error and KL-based loss as class distribution loss.

Authors of [43] analyze few-shot learning in the field of image colorization. Their model Memo-Painter has a standard cGAN architecture composed of a U-net generator and a binary discriminator, but with a memory augmented network that helps capture and remember rare object instances. Queries for the memory network are constructed by passing the input image through a pretrained ResNet18 network, while color distributions and dominant RGB color values are stored in the network. A novel threshold triplet loss is also introduced.

The approach described in [44] utilizes guided image colorization, meaning that automatically chosen reference images help aid the colorization process. The model consists of a reference component matching module and a double-channel colorization module. The reference module uses a pretrained ResNet18 network to find the color channels’ most appropriate reference feature vectors. These reference feature vectors are then used as an additional condition besides the grayscale image to the two GANs, one for each channel. Both of the GANs follow Pix2Pix architecture.

This is the first paper that implements and train a conditional generative adversarial network for image colorization as a proxy task. The proposed model is inspired by Pix2Pix [33], a network that provides a general solution to many image-to-image translation problems, such as, day to night, edges to photo, aerial to map, black and white to color, etc. The architecture of the traditional generative adversarial network needs to be modified for it to be applicable to the image colorization problem. Instead of a noise vector, the generator is altered to accept a grayscale image as input, or in other words, the generator is conditioned on the grayscale image. The discriminator is also modified to accept a grayscale image alongside an actual or predicted image. This type of network is called a conditional generative adversarial network [17] and a general outline of its architecture is shown in Figure 3.

The generator is based on the U-Net model [45], initially designed for image segmentation. U-Net is a convolutional neural network that has an encoder–decoder structure. The input images are first gradually downsampled through a series of convolutions until they reach a bottleneck layer, which contains a condensed learned representation of the images. After the bottleneck, the images are progressively upsampled until they reach the desired output dimensions. Skip connections that connect outputs from the downsampling path with the upsampling path are also added. They assist the flow of low-level information through the network, as the bottleneck layer prevents this. Both the encoder and decoder are made up of seven convolutional blocks. The decoder uses dropout to avoid overfitting and add diversity to the generated images. All activations are ReLU or LeakyReLU, except the last one, which is Tanh.

The discriminator is called a PatchGAN and it is also a convolutional neural network. Typically, discriminators give one probability for the whole image that tells us if that image is real or fake. In contrast to that, PatchGAN splits the image into NxN patches and outputs a matrix of probabilities for each patch. This allows for getting more informative feedback from the discriminator. One part of the image can be considered realistic, while another part may need improvement. The discriminator is made up of four convolutional blocks. All activations are LeakyReLU, except the last one, which is sigmoid. The receptive field of the PatchGAN is 70 × 70 pixels, following the best practices from [33].

### 3.3. Image Colorization Model

This subsection explains the learning procedure and details of training the previously described model for image colorization as a proxy task.

#### 3.3.1. Dataset

Image data used for training the image colorization model comes from the COCO (Common Objects in Context) [8] 2014 training set. Even though the COCO dataset is designed for computer vision tasks, such as classification and detection, and comes with supervised labels, none of the provided labels were used. The subset used for training contains 83 thousand images from 80 different categories, including people, vehicles, animals, food, outdoor objects, and more. The COCO dataset provides images with everyday scenes. Therefore, it is appropriate for training a colorization model with general world knowledge. Even though most recent colorization models are trained on ImageNet, consisting of 1.2 million training images, we train on a much smaller dataset because of hardware and time limitations. This choice makes comparison with other models challenging. However, we can still measure the improvement by comparing the results to the model that does not use colorization as a proxy task. For testing the cGAN colorization model, 4000 randomly chosen images from the COCO 2014 test dataset were used.

#### 3.3.2. Color Space

As previously mentioned, a luminance–chrominance color space is needed for the image colorization task to separate the intensity from the color information. The CIELAB (Lab) is one such color space used to describe all visible colors by the human eye. It was created to represent color changes in the same way as humans do. This means that a numeric change corresponds to a similar perceived difference in color. The space has little correlation between its three components. The L component stands for perceptual lightness with range [0, 100], meaning that it is the grayscale element. The A component represents the color position between red and green, while the B component represents the color position between blue and yellow; both components have ranges [−128, 127]. Before entering the model, all channels are normalized in the range [−1, 1]. The L channel is used as an input to the model, while A and B channels are the target values.

#### 3.3.3. Objective Function

The objective functions used for training conditional generartive adversarial networks is as follows [33,34]:(1)$$\min\max V\left(G,D\right)=E_{x,y∼P_{data}\left(x\right)}\left[\log D\left(x|y\right)\right]+E_{y,z∼P_{z}\left(z\right)}\left[\log\left(1-D\left(G\left(z|y\right)\right)\right)\right]$$

The generator *G* tries to minimize the objective function while the discriminator *D* tries to maximize it, where *x* is the input grayscale image and *y* is the output color channels.
(2)$$L_{L1}\left(G\right)=E_{x,y,z}{[∥y-G\left(x,z\right)∥}_{1}]$$

Mean absolute error (*L*1 loss) is also included to help generate realistic images with a structure close to the original image. This loss is treated as a regularizing term, and it is weighted with the hyperparameter lambda. With the *L*1 loss added, the final objective function is as follows:(3)$$G^{*}=\arg\min\max L_{cGAN}\left(G,D\right)+\lambda L_{L1}\left(G\right)$$

Instead of using a noise vector to add diversity to the results, [33] suggests only using dropout layers for this purpose, as the network learned to ignore the noise. This dropout is also utilized during the inference mode of the model.

#### 3.3.4. Metrics for Evaluating Image Colorization Methods

Objectively comparing and evaluating different methods for image colorization is inherently challenging as there are subjective aspects if human evaluation is performed (e.g., human eye perception, biological differences of evaluators, varying display quality, etc.). Fortunately, in computer vision tasks, there are two commonly used objective metrics that employed for such evaluation: the PSNR (peak signal-to-noise ratio) and SSIM (structural similarity index measure). PSNR is a metric used to estimate the quality of image reconstruction. The formula for calculating the PSNR score is as follows [46]:(4)$$\mathrm{MSE}=\frac{1}{H*W}\sum_{i=1}^{H}\sum_{j=1}^{W}\left[X\left(i,j\right)-Y\left(i,j\right)\right]$$
(5)$$\mathrm{PSNR}=10\log_{10}\left(\frac{{\left(2^{n}-1\right)}^{2}}{MSE}\right)$$
where *H* and *W* represent the height and width of the image, *i* and *j* are the pixel coordinates, *n* is the bit of pixels, and MSE stands for mean squares error. SSIM on the other hand, is a similarity metric that takes into consideration the luminance, contrast, and structural changes between two images. It was made to more consistent with human eye perception of similarity, and, therefore, it is an improvement on the MSE and PSNR metrics. The formula for luminance comparison is as follows:(6)$$l\left(x,y\right)=\frac{2\mu_{x}\mu_{y}+C_{1}}{\mu_{x}^{2}+\mu_{y}^{2}+C_{1}}$$

The formula for contrast comparison is as follows:(7)$$c\left(x,y\right)=\frac{2\delta_{x}\delta_{y}+C_{2}}{\delta_{x}^{2}+\delta_{y}^{2}+C_{2}}$$

The formula for structure comparison is as follows:(8)$$s\left(x,y\right)=\frac{\delta_{xy}+C_{3}}{\delta_{x}\delta_{y}+C_{3}}$$

The final SSIM formula is as follows [46]:(9)$$SSIM\left(x,y\right)=\frac{\left(2\mu_{x}\mu_{y}+C_{1}\right)\left(2\delta_{xy}+C_{2}\right)}{\left(\mu_{x}^{2}+\mu_{y}^{2}+C_{1}\right)\left(\delta_{x}^{2}+\delta_{y}^{2}+C_{2}\right)}$$
where $\mu_{x}$ and $\mu_{y}$ are the average values of the real and colored images, respectively, $\delta_{x}^{2}$ and $\delta_{y}^{2}$ are the variances of real and colored images and $\delta_{xy}$ is their covariance. $C_{1}$, $C_{2}$, $C_{3}$ are constants used to avoid errors when the denominator is 0.

A higher PSNR or SSIM score means better image reconstruction. Considering that the SSIM score of two identical images would be 1, an average SSIM score of 0.85 on the test set shows that the colorization model has learned to accurately colorize previously unseen images.

### 3.4. Transfer Learning to Downstream Tasks

In order to evaluate the quality and usefulness of the learned features through self-supervised proxy tasks, the learned parameters are usually transferred to other downstream tasks such as classification, object detection, and segmentation [16,30,31,32]. The data flow diagram of training the colorization model and transferring its weights to downstream tasks is shown in Figure 2. If the models have learned general visual features, then using the parameters from the pretrained models would be a good starting point for other similar computer vision tasks. After being initialized with the learned parameters, the models are then fine-tuned to the downstream tasks. The fine-tuning should require less training time and a smaller dataset than a network trained from scratch, as the model already comes with knowledge. In addition, the evaluated performance of these downstream tasks demonstrates the generalization ability of the learned features.

This paper uses the parameters learned through self-supervised colorization and transfers them to two other tasks, multilabel image classification, and semantic segmentation. We then examine if using colorization as an initialization for the networks results in a better performance than training the models from scratch.

#### 3.4.1. Dataset

The data used for fine-tuning and evaluating the downstream tasks of multilabel classification model and semantic segmentation comes from the Pascal VOC 2012 dataset [9]. There are 20 classes total, including airplane, bicycle, bird, boat, bottle, bus, car, and more and some exemplary images are shown in Figure 4. It is essential to note that no Pascal VOC images were used for training and testing the image colorization model.

#### 3.4.2. Multilabel Image Classification Model

In the context of computer vision, multilabel classification can assign multiple classes to a single image. It is a variant of image classification that allows multiple objects to be detected in one image.

For this task, the number of training/fine-tuning images is 11,987, the number of validation images is 3425, and 1713 images were used for testing. We made sure that each data split had the same distribution of classes.

Each image is converted into the Lab colorspace and only the L grayscale channel is used as input to the model. This is done in order to be able to reuse the parameters and architecture of the colorization model. As stated in [16], using only the grayscale version of the image as opposed to the color version makes almost no difference in the performance of the models. The input images are also normalized in the range [−1, 1]. The target values for the multilabel classification model are vectors of 20 elements, where 0 is assigned if the class is not present or 1 if the class is present on the image.

The architecture used for the multilabel classification model is shown in Figure 5. We start building the multilabel classification model by reusing the encoder part of the generator from the image colorization model and discarding the decoder part and the discriminator. The encoder should have learned to extract important features from the training images. It ends with the bottleneck layer, where all knowledge is condensed. After the encoder, new layers are added that are suitable for classification. First, the bottleneck layer is flattened, after which two dense layers of 4096 neurons are added, similar to a VGG-16 classificator. Both dense layers are followed by a ReLU activation function, after which dropout layers with a 0.5 rate are added to prevent overfitting. The model ends with another dense layer of 20 nodes, as we have 20 different classes. In the end, a sigmoid activation function is used to give a prediction probability for each class. The outline of the model’s architecture can be seen in Figure 5.

#### 3.4.3. Semantic Segmentation Model

Semantic segmentation is the task of assigning semantic labels to each pixel of an image. More specifically, each pixel is given a class of the object that it belongs to. Because every pixel is given a prediction, this is commonly referred to as dense prediction. Semantic segmentation does not differentiate between multiple instances of the same object. Semantic segmentation models are used in many different tasks, such as autonomous driving, virtual try-on, visual search, medical image analysis, and more.

All data for fine-tuning and testing the semantic segmentation model also comes from the Pascal VOC 2012 dataset. The number of images used for training is 2.039, the number of validation images is 577, and 297 images were used for testing. The same 20 classes of objects are present on the images as in the classification dataset, with the addition of a background class. Each image in the dataset comes with a mask, where the objects on the image are segmented, each class with a different color or shade. For example, the background class is black, and the bicycles are green. Example images and their masks can be seen in Figure 4. Every image is preprocessed in the same way as in the classification dataset. All images with their masks were randomly cropped to fit 256 × 256 dimensions. The target values have a shape of (256, 256, 21), where each pixel is a one-hot vector, with one meaning that the pixel belongs to the class.

The architecture used for the multilabel classification model is shown in Figure 6. Here, we reuse the whole generator from the image colorization model, including the encoder, the bottleneck, the decoder, and the skip-connections. Only the output layer is different to accommodate the different shape of the output matrix. The model ends with a softmax activation function, as each pixel can belong to only one class. This type of architecture, in a form of U-net, is common for image segmentation models [45,47]. The general outline of the model’s architecture can be seen in Figure 6. Because the image colorization task and the image segmentation task are similar, and they give pixelwise predictions, we expect the transferred features to be easily fine-tuned.

## 4. Results

### 4.1. Image Colorization

The colorization model was trained for 20 epochs on an Nvidia GeForce GTX 1660 SUPER graphics card. The whole training process took around 24 h. All images were resized to 256 × 256 dimensions. They were fed to the model in batches of 32. Adam was used as an optimizer with a learning rate of 0.002. The code for training and evaluating the image colorization GAN is publicly available in the following repository [48], and the code for transferring the gained knowledge to other downstream tasks is available at [49]. The learning progress was monitored and can be seen on Figure 7.

The cGAN colorization model was evaluated on the COCO test dataset using the PSNR (peak signal-to-noise ratio) and SSIM (structural similarity index measure) metrics. PSNR is a metric used to estimate the quality of image reconstruction. Higher PSNR or SSIM scores denote better image reconstruction. Table 3 shows the average, minimum, and maximum values of the PSNR and SSIM scores across all testing images. Considering that the SSIM score of two identical images would be 1, an average SSIM score of 0.85 on the test set shows that the colorization model has learned to accurately colorize previously unseen images.

### 4.2. Multilabel Classification

Two variants of the multilabel classification model were trained. The first one was initialized from a Gaussian distribution with mean 0 and standard deviation 0.02, meaning it was trained from scratch and it will be our baseline. The second model was initialized with the learned weights from the image colorization model, except the added classifier’s weights. Both models use binary cross-entropy loss, an Adam optimizer, and a batch size of 32. The first model was trained for four epochs with a learning rate of $3\times {\mathrm{10}}^{-4}$. The second model was trained in two phases. For the first three epochs, the colorization weights were frozen, and a learning rate of $3\times {\mathrm{10}}^{-4}$ was used. For the second phase of one epoch, all weights were unfrozen, and a lower learning rate of $5\times {\mathrm{10}}^{-5}$ was used. The two-phase training is done so that a large learning rate does not immediately destroy the learned weights. Both models were trained using GPU-powered Google Colab notebooks.

After being trained, the classification and segmentation models were evaluated using the Pascal test dataset. Table 4 shows the accuracies of the multi-label classification models. The model which uses colorization pre-trained initialization achieves around a 5% improvement over the model trained from scratch. This shows that the features learned from the colorization model were successfully transferred to the task of multi-label classification. The features were useful, as similar objects are present in the COCO and Pascal datasets. One thing to note is that our goal here was not to create the most accurate classification model for the Pascal dataset, rather to illustrate the importance of the colorization pre-training with the relative improvement.

### 4.3. Semantic Segmentation

Two versions of the image segmentation model were also trained. One starts from random initialization and the other starts from image colorization pretrained weights. Both models use categorical cross-entropy as a loss function, an Adam optimizer and a batch size of 16. The models were trained for six epochs. The pretrained model was trained in two phases, for one epoch, the weights were frozen, while for the rest of the five epochs they were unfrozen. The training losses of both models can be seen in Figure 8 (right graph), and here again, one can notice that the second model starts with an advantage. Both models were trained using GPU-powered Google Colab notebooks.

The semantic segmentation models were evaluated using the metric mean intersection over union (mIU). Intersection over union (IoU) is a method that quantifies the percentage of overlap between the actual and predicted masks, and it is also called Jaccard index. This score is calculated for each class separately and then averaged over all classes to create the mean intersection over union (mIU) score used to asses the performance of the semantic segmentation models. From Table 4, it is clear that the model that starts from colorization pretrained weights achieves a better mIU score and improves upon the model trained from scratch with around 2.5%. This difference is slighter than the one between the classification models. We assume this is because the segmentation dataset is significantly smaller than the classification dataset. Again, our goal was not to create the best performing segmentation model, but simply to prove that the colorization pretraining increases the performance of other downstream tasks. To further improve the models, larger datatsets for pretraining and fine-tuning are needed, as the Pascal VOC tasks are complex and require knowledge of many different objects.

## 5. Discussion

Inspired by the recent advances in the field of image colorization by using conditional generative adversarial networks, we extended the existing research by exploring ways to transfer the gained general knowledge to other tasks of different domains. We suggested what parts and layers of the image colorization GAN can be reused for two other tasks and how to extend the architecture with other layers specific to those domains. Choices for different hyperparameters and training strategies were also shared.

The presented evaluation shows that reusing parts of the GAN architecture and weights for other downstream tasks resulted in their improved performance. The training losses of both models can be seen in Figure 8 (left graph). It can be noticed that the loss from the model initialized with the colorization weights decreases much more quickly than the model trained from scratch. Even after the first epoch the pretrained model shows an advantage compared to the other model, which proves that the network comes with prelearned useful features.

In order to enable a fair comparison to previous self-supervised learning methods, they need to be trained and evaluated on the same datasets. Therefore, the colorization pretraining can be extended to use the much bigger ImageNet dataset in the future, as most other image colorization papers use this dataset. Another future direction is to include more downstream tasks for evaluation and thoroughly investigate the generalization ability of the learned weights through colorization. As shown in this research, such transfer learning approaches are promising and could further boost the performance of models for image generation or segmentation of specific objects and regions. There are plenty of such problems in various areas, including the medical domain where many applications require precise segmentation of various tissues (e.g., skin lesion segmentation [50], lung nodule segmentation [51], etc.) in magnetic resonance imaging (MRI) and computerized tomography (CT) scans. In such scenarios, the datasets will likely increase in size and variety, which in turn will entail the use of appropriate architectures for Big Data processing and learning [52,53].

## 6. Conclusions

In this paper, we explored image colorization as a proxy task for visual understanding for the first time. Current and relevant research was first analyzed and compared. Then, conditional generative adversarial networks were presented as a way to train an image colorizer for the purpose of visual feature learning. Finally, the gained knowledge was transferred to two other downstream tasks, namely multilabel image classification and semantic segmentation. The experiments successfully demonstrate that the models starting from the pretrained colorization weights showed better performance than the models trained from scratch without such initialization. Therefore, we can conclude that self-supervised visual feature learning by using cGANs on the image colorization task is worthy of further exploration and has the potential to come close to the traditional supervised pretraining.

## Figures and Tables

**Figure 1 sensors-22-01599-f001:**
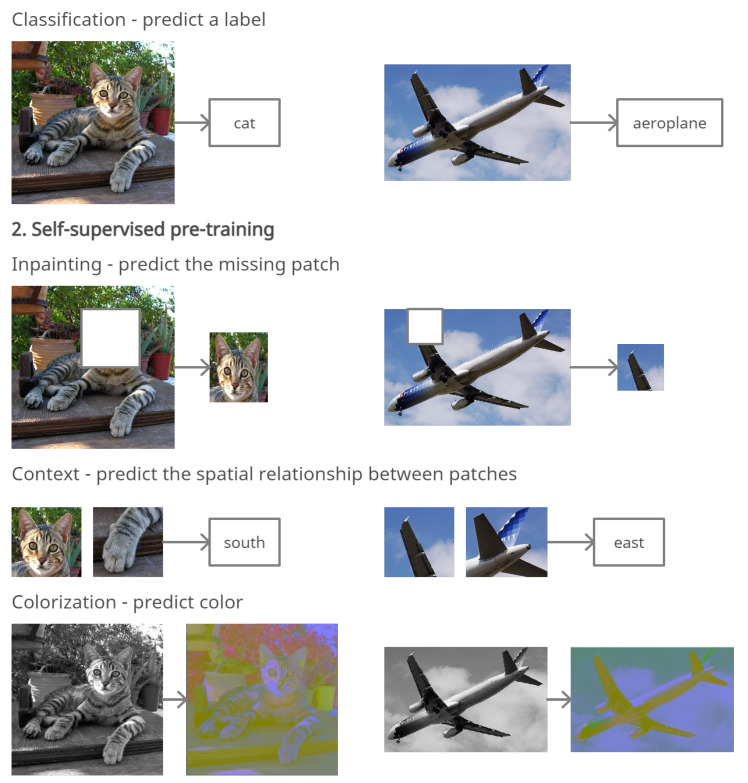
Supervised and self-supervised methods for learning visual features.

**Figure 2 sensors-22-01599-f002:**
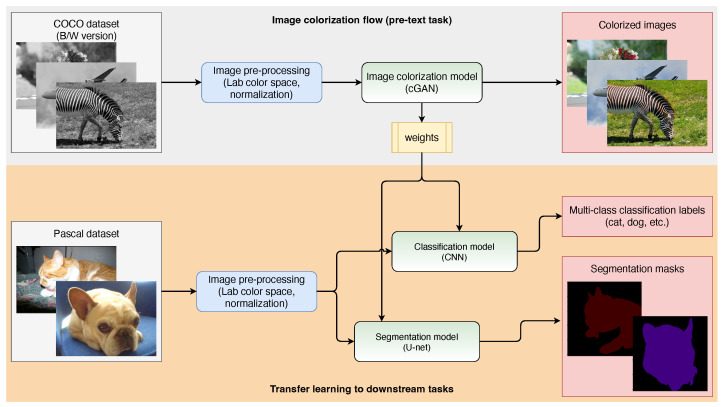
Data flow diagram of training the colorization model and transferring its weights to downstream tasks.

**Figure 3 sensors-22-01599-f003:**
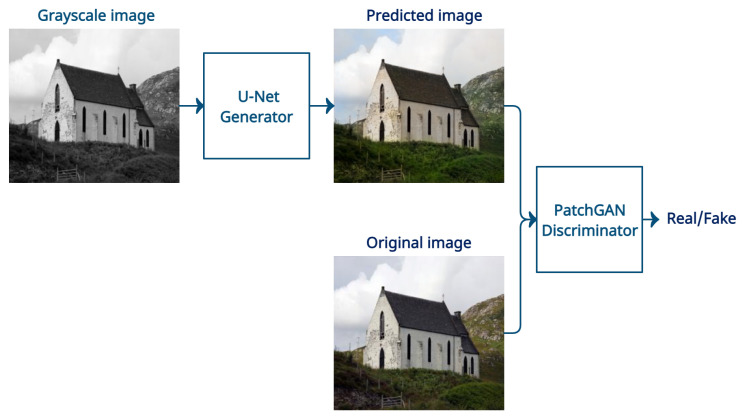
Architecture of a conditional generative adversarial network for image colorization.

**Figure 4 sensors-22-01599-f004:**
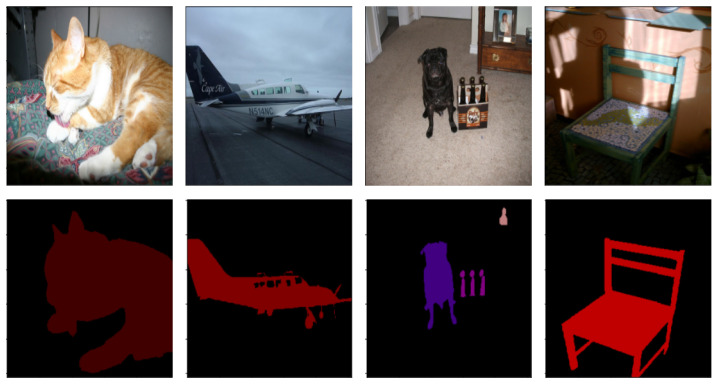
Example images and their masks from the Pascal VOC segmentation dataset.

**Figure 5 sensors-22-01599-f005:**
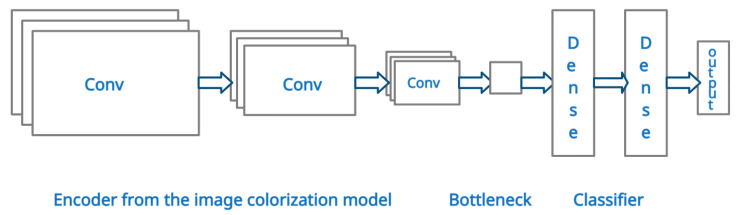
Architecture of the multilabel classification model.

**Figure 6 sensors-22-01599-f006:**
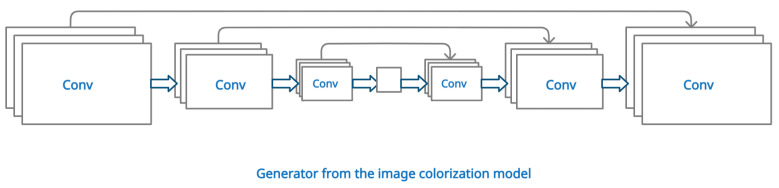
Architecture of the semantic segmentation model.

**Figure 7 sensors-22-01599-f007:**
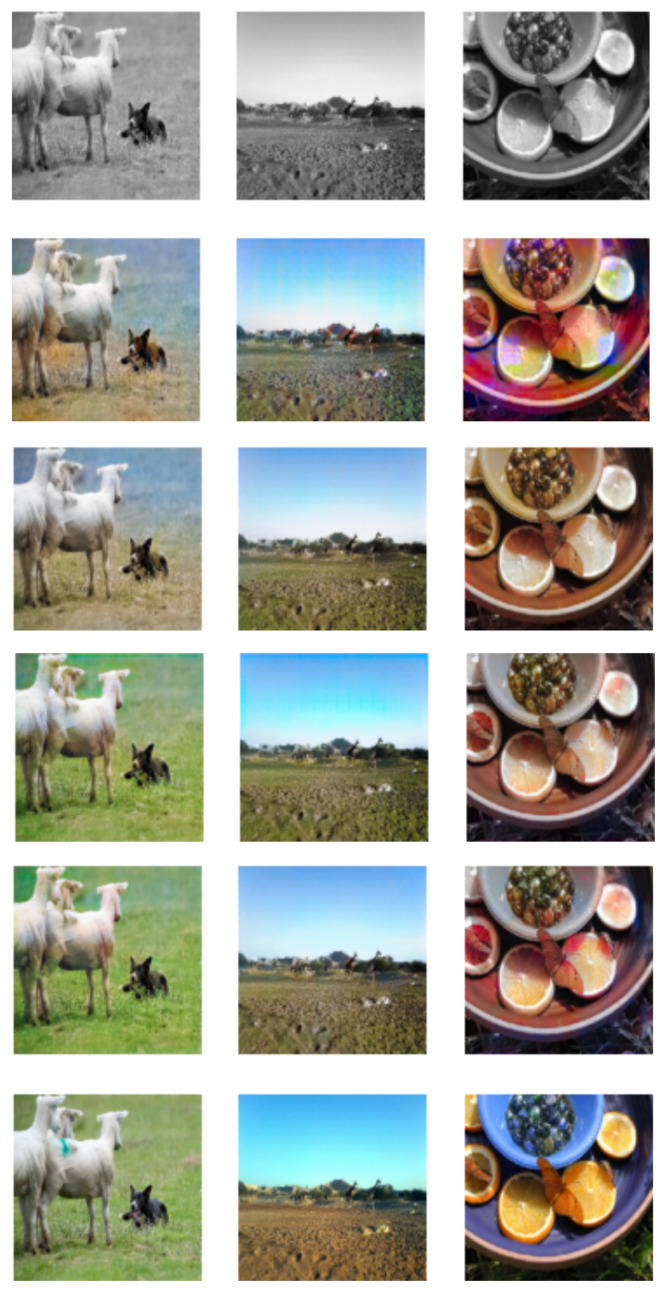
Learning progress of the image colorization model. On the first row are the grayscale input images. Next are the learned colorizations after 1, 5, 15, and 20 epochs. On the last row are the original color images.

**Figure 8 sensors-22-01599-f008:**
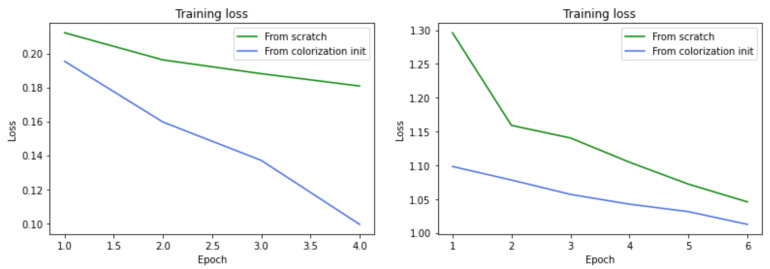
Training losses of the multi-label classification models (**left**) and semantic segmentation models (**right**).

**Table 1 sensors-22-01599-t001:** Comparison of previous research papers that investigate image colorization as a proxy task.

Paper	Model Base	Pretrain Dataset	Fine-Tune Dataset
[16]	AlexNet	ImageNet	PASCAL VOC
[30]	VGG-16	ImageNet	PASCAL VOC
[31]	AlexNet, VGG-16, ResNet-152	ImageNet, Places	PASCAL VOC
[32]	Cross-channel autoencoder	ImageNet, Places	PASCAL VOC

**Table 2 sensors-22-01599-t002:** Comparison of the best results of previous research on the Pascal VOC dataset for classification, segmentation, and object detection downstream tasks. The bolded result in each column denotes the best result for that task.

Paper	Classification (mAP%)	Segmentation (mIU%)	Detection (mAP%)
[16]	65.9	35.6	**47.9**
[30]	/	50.2	/
[31]	**77.3**	**60.0**	/
[32]	67.1	36.0	46.7

**Table 3 sensors-22-01599-t003:** Evaluation of the cGAN colorization model on the COCO test dataset.

Metric	Average	Min	Max
PSNR	20.94	8.82	42.61
SSIM	0.85	0.31	0.99

**Table 4 sensors-22-01599-t004:** Evaluation of the multilabel classification and semantic segmentation models on the Pascal VOC test dataset.

Model Initialization	Classification (Acc)	Segmentation (mIU)
Baseline	47.18%	44.66%
Colorization pre-training	52.83%	47.07%

## Data Availability

The source code developed in this study is available at [48,49]. The datasets used in this study are available at: ImageNet [7], COCO [8], and Pascal VOC [9].

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
