# Peer review of "GAN-Based Image Colorization for Self-Supervised Visual Feature Learning"

_sensors, 2022, doi:10.3390/s22041599_

Round 1
Reviewer 1 Report
This paper proposes to use GAN for image colorization by self-supervised learning and then guide downstream tasks. Experimental results on multi-label image classification and semantic segmentation demonstrate the effectiveness of the proposed model. Some comments are listed as follows:
- The motivations of adopting features from colorization for classification and segmentation should be addressed.
- Can the proposed model be transferred to other downstream tasks?
- It is suggested to compare with some state-of-the-art methods. Both quantitative and qualitative results would be helpful to further verify the proposed method.
- To make a better understanding for readers, self-supervised learning in other computer vision and image processing should be reviewed, including anomaly detection (CutPaste: Self-Supervised Learning for Anomaly Detection and Localization), image deraining (Unsupervised single image deraining with self-supervised constraints), etc.
- Except for PSNR/SSIM, some naturalness metrics such as NIQE could be considered for evaluation.
- It would be better to make the code publicly available.
Reviewer 2 Report
This work demonstrates an initial application of GAN for self-supervised visual feature learning through image colorization.
The introduction is informative, and the objectives are clear. Methods are well described and results seem to be reliable. The paper also includes an interesting discussion and a well-written conclusion.
I have one comment: If applicable, I suggest the authors provide a training and validation accuracy diagram for both cases of multi-label classification models and semantic segmentation models for a higher number of epochs (preferably higher than 50). It will be useful to describe the overfitting and underfitting conditions through such a diagram as well.
Reviewer 3 Report
All figures and tables should have a source and should be referenced in the text. I think that tables should be given later in the text after referenced.
Related work review should be more content related. Would be interested to the reader to have info about other works results, not only just the methods they used and the purpose. Some examples are provided but some more would enrich this paper.
In Discussion part would appreciate the comparison with other research results. What did you achieved better comparing to others with examples.
In Conclusions part some future research directions and weaknesses of your analysis should be presented.
Reviewer 4 Report
Dear Authors,
the manuscript presented for evaluation is very interesting and valuable for the presented field. Overall, the work is well written and correctly arranged, the parts are well edited, the methodology of the work is properly presented, adequate literature has been used, and it forms a coherent whole. Despite the great care of the work, I found a few points that should be improved or clarified.
Notes to the manuscript:
- In the Intoduction section, the text reference to Figure 1 is missing.
- Page 3 the caption of Table 1 should be placed above the table, a similar note applies to the other tables in the manuscript (tables 2, 3 and 4).
- Figure 3 should be moved to the next paragraph (line 220).
- Figure 7 should be better placed in the text, there are too large gaps before and after the drawing.
- Chapter 5. Discussion is too short and does not relate to other works that were presented, for example, in the earlier parts of the article. At this point, there should be both a discussion of one's own results and their comparison against the background of available works in the literature on the issue. Therefore, I am asking you to strengthen this part of the article.
Apart from these comments, I have no major objections and I believe that the work is suitable for the Sensors journal.
Thank you!
Round 2
Reviewer 1 Report
The paper can be accepted in its current form.
Author Response
Thank you for the valuable comments. The language in the paper has been further improved.